# Forecasting Commodity Market Synchronization with Commodity Currencies: A Network-Based Approach

**DOI:** 10.3390/e25040562

**Published:** 2023-03-25

**Authors:** Nicolas S. Magner, Nicolás Hardy, Jaime Lavin, Tiago Ferreira

**Affiliations:** 1Facultad de Administración y Economía, Universidad Diego Portales, Santiago 8370109, Chile; 2Escuela de Negocios, Universidad Adolfo Ibáñez, Peñalolén 7941169, Chile; 3Facultad de Economía y Negocios, Universidad Alberto Hurtado, Santiago 6500620, Chile

**Keywords:** commodity markets, returns synchronization, networks analysis, forecasting models, commodity currencies

## Abstract

This paper shows that some commodity currencies (from Chile, Iceland, Norway, South Africa, Australia, Canada, and New Zealand) predict the synchronization of metals and energy commodities. This relationship links the present-value theory for exchange rates and its connection with commodity export economies’ fundamentals, where prospective commodity price fluctuations affect exchange rates. Predicting commodity market return synchronization is critical for dealing with systemic risk, market efficiency, and financial stability since synchronization reduces the benefits of diversification and increases the probability of contagion in financial markets during economic and financial crises. Using network methods coupled with in-sample and out-of-sample econometrics models, we find evidence that a fall in the return of commodity-currencies (dollar appreciation) predicts an increase in commodity market synchronization and, consequently, in commodity market systemic risk. This discovery is consistent with a transitive capacity phenomenon, suggesting that commodity currencies have a predictive ability over commodities that extend beyond the commodity bundle that a country produces. The latter behavior would be exacerbated by the high financialization of commodities and strong co-movement of commodity markets. Our paper is part of a vigorously growing literature that has recently measured and predicted systemic risk caused by synchronization, combining a complex systems perspective and financial network analysis.

## 1. Introduction

Financial integration has recently increased due to significant economic openness and extensive commercial integration among markets and countries. Capital markets are not the exception since a broad interconnectedness coupled with significant size and complexity transformed them into extensive complex systems. As a result, financial markets become highly interconnected with cross-border links and interdependencies among their constituencies, which easily amplify and transform economic, financial, or sanitary shocks into global and comprehensive extended events [1,2,3].

In this new context, synchronization of the returns of financial assets becomes critical. Synchronization impacts systemic risk and the aggregated behavior of capital markets, making it a relevant factor in assessing the risk of financial contagion, the stability of markets, and the cost and effectiveness of diversification as a tool for risk management. The latest crisis episodes in the capital markets, such as the Subprime Crisis of 2007–8 and the COVID-19 outbreak in 2020, show that investors desperately seek risk-free assets. Consequently, risk appetite fades, provoking the liquidity of risky assets to practically disappear. Consequently, episodes of simultaneous and widespread declines in the prices of financial assets in different markets and countries are easily observable during the high synchronization of returns [4].

The literature has studied the synchronization phenomenon extensively. Works on interconnectedness show its dual impact on systemic risk; on the one hand, it could improve financial robustness when it contributes to absorbing shocks. However, on the other hand, it could also generate contagion when shocks spread among the components of a financial system. For example, evidence shows that during high-volatility stock market periods, the correlation between stocks rises, increasing the synchronization of returns, threatening the benefits generated by diversification, and lessening the financial system’s stability [5,6].

Understanding the synchronization of returns is highly relevant in financial markets because, under episodes of contagion, market volatility spurs, changing the correlation coefficients among financial assets [7]. Moreover, economic structural similarities between the structure of national and regional economies, coupled with global factors, explain markets’ co-movement, which eventually generates financial contagion on a large scale [8]. For example, at the stock market level, Lavin, Valle, and Magner [4] find that during global equity synchronization episodes, regional stock markets react in the same direction as global markets, but with different sensitivity to a shock that spurs global synchronization.

Synchronization behavior is not exclusive to equity or bond markets; there is also rising interconnectedness in international commodity markets. After the global financial crisis of 2008, there was a significant increase in connectedness risk in global commodity markets, which depicts a growing interdependence among them [9]. This new scenario of larger interconnectedness among commodities imposes new challenges on investors mandated to manage portfolios that include commodities and policymakers required to anticipate periods of extreme price instability [10]. Moreover, evidence indicates that interconnections among capital markets vary over time, an uneven phenomenon among countries and regions [11].

Commodity prices have experienced pronounced shifts over the last few years. The literature notes that price dynamics have changed since 2000, particularly around the 2007–8 Global Financial Crisis. The growing path of cross-commodity correlations and the correlation of commodity prices with the prices of other financial asset classes have attracted greater interest. For example, Cheng and Xiong [9] found that correlations passed from −0.2 to 0.2 in 2004 to a peak of 0.7 in 2008. Even across sectors, they document that commodity prices have tended to move together as the same class since the 2000s. Similarly, correlations of commodity prices with prices in other asset classes trended upward from 2004 to 2008, increasing significantly since 2008. These authors also point out that this latter behavior is consistent with the notion that commodity prices have shared a typical boom and bust cycle due to the growing financialization of commodity markets [9].

Evidence shows that commodities behave in a correlated way. For instance, Pincheira and Hardy [12] found that copper correlates with aluminum, lead, nickel, tin, and zinc. They argue that this phenomenon is because all these metals have industrial manufacturing uses and global consumption. Additionally, all these commodities are traded daily on the London Metal Exchange (LME), the New York Mercantile Exchange (NYMEX), and the Chicago Mercantile Exchange (COMEX), where they are connected by commodities indexes and derivatives contracts such as options and futures. These market connections and economic and financial elements cause co-movement behavior and synchronization phenomenon [13].

Given the importance of exchange rates to future economic expectations, a large body of literature assesses commodity currencies’ predictive ability. In a seminal paper, Chen, Rogoff, and Rossi [14] observed that the currencies of five commodity-exporting economies (Australia, Canada, Chile, New Zealand, and South Africa) can predict the prices of the commodities they export. In particular, they built commodity indices based on their export baskets. Chen, Rogoff, and Rossi [15] repeated the exercise with agricultural commodities, finding quite a bit of predictability. These works show the pervasive relationship between commodity currencies and their prospective ability on the price of commodities.

The economic mechanism that explains the predictive capacity of commodity currencies relates to the present value model for determining exchange rates [16,17]. This literature, named the “Commodity Currency Hypothesis (CCH),” has attracted the attention of practitioners and scholars worldwide. Gargano and Timmermann [18] found evidence of CCH using the Australian dollar and the Indian rupee. According to Pincheira and Hardy [12], the Chilean exchange rate can predict the return of aluminum, copper, lead, nickel, tin, and zinc. The authors explain their results based on plausible explanations that may rely on strong co-movement of commodity returns, which is a novel aspect of this result. Chile only produces copper on this list, so it is reasonable to expect that it should only be able to predict copper returns.

Similarly, Pincheira and Hardy [19] observed that six commodity currencies (Australia, Canada, Chile, Iceland, New Zealand, and South Africa) can predict aluminum quite well, even though some of these countries are not even aluminum producers or exporters. In this way, Pincheira, Hardy, and Betancor [20] showed that the Chilean currency can predict fuel prices, although Chile does not produce fuels. Altogether, these findings point to a transitive capacity phenomenon, in which commodity currencies have the predictive ability over commodities that extend beyond the commodity bundle that a country produces.

Predictive ability could go beyond the present value relationship derived from a leverage effect, a dollar effect, and the relationship between the currency and its fundamentals. Groen and Pesenti [21] use ten alternative commodity indices to those of Chen Rogoff, and Rossi [14], finding results that only partially support the CCH. Finally, Bork, Kaltwasser, and Sercu [22,23] and Lof and Nyberg [22,23] use the same data as Chen Rogoff, and Rossi [14] and give different arguments to support that there is no predictive relationship. As these previous works show, there is still room for a deeper understanding of the predictive capacity of commodity currencies on the behavior of commodity prices.

Moreover, to the best of our understanding, there is no evidence regarding the predictive abilities of these currencies on the synchronization behavior of global commodity markets and, consequently, on their possible impact on the stability of commodity markets. Accordingly, we contribute to closing a gap in the literature to gain a deeper understanding and assessment of commodity markets’ behavior and impact in terms of their prices’ co-movements. Because they are an essential component of the global economy, it is critical to comprehend how risk reallocation and information transmission from commodity markets due to synchronization shocks affect the real economy and global financial markets.

As a result, this study demonstrates commodity currency predictive ability on the synchronization behavior of commodity market returns, considering base metals, precious metals, and energy commodities. The network comprises the following commodities: aluminum, copper, lead, nickel, tin, zinc, gold, platinum, silver, palladium, gasoline, wti, brent, heating oil, and natural gas. For this, we used asset correlation network methods (Minimum Spanning Tree Length, MSTL [24]) to measure commodity synchronization and standard econometric forecasting models to test the predictive ability of commodity currencies on commodity market synchronization.

We conjecture that this association is in line with the commodity currency hypothesis (CCH), which points out that the currencies of highly exporting economies can anticipate movements of certain commodity prices. As a base for the above, commodity currencies are strongly forward-looking since they embody information about future movements in the commodity markets that cannot easily be captured by simple time series models [25].

Our analysis covers two aspects: the relationship between the network of commodities and commodities-currencies and their relationship over time. First, our findings agree with [12,15,19,26], who reported a robust predictive ability of commodity currencies over commodity returns. Second, our results are consistent with findings from Magner, Lavin, Valle, and Hardy [27] and Lavin, Valle, and Magner [28], who pointed out that stock networks tighten up in times of crisis during periods of lower returns and higher volatility.

Additionally, our out-of-sample tests reported some striking results for predictability. Specifically, our models statistically outperformed the respective benchmarks in 42% of the exercises. Moreover, Norway’s currency, for instance, outperformed most benchmark models across all estimation window definitions, reaching significance in 20 of 21 exercises, followed by models using Canada’s currency, which reached significance in 13 of 21 exercises, and models using Australia’s currency, which achieved significance in 10 of 21 exercises. Additionally, regarding the behavior of the stability of the previous relationships through time, our results are consistent with previous literature (see [29]), indicating that the predictive ability of financial assets seems to be unstable, with sharp swings in magnitude and significance when considering different estimation/forecasting window periods.

Finally, we implemented a directional accuracy test to examine the accuracy of commodity currencies in anticipating changes in direction of commodity market synchronization. All of our exercises were statistically significant, with a 1% level. Furthermore, the average hit rate was about 74%, which is in stark contrast to a “pure luck” 50% probability. Based on the evidence above, our core models are effective instruments for forecasting commodity market synchronization and, as a result, their financial behavior and stability performance.

Our work is important in several aspects. First, at the macroeconomic level, it allows for predicting the synchronization behavior of the international commodity markets. It is helpful for regulators in monitoring international risk contagion and its impact on financial stability. The argument is that an increase in the probability of synchronization is directly related to an increase in the risk of financial contagion threatening the financial stability of both the financial and commodity markets.

Second, at the investor level, our work helps to anticipate commodity market synchronization episodes using commodity currencies as input. This anticipation ability will allow investors and market participants to adapt their portfolios to optimize diversification and manage financial risk better.

Finally, our work contributes to the growing literature that, in recent years, measures and studies systemic risk by applying complex system insights and financial network analysis in conjunction with advanced forecasting time-series models. We contribute to a better assessment of the impact of commodity markets.

Our results have practical implications for practitioners dealing with market synchronization risk:i.Synchronization is crucial for systemic risk and financial stability in economic and financial crises. When synchronization in the financial market increases, it reduces the benefits of diversification and increases the probability of contagion.ii.As the benefits of international diversification diminish, risk management strategies based on investing abroad and combining multiple financial asset classes with commodities require a proper redesign to include this risk factor in the decision-making criteria.iii.The emergence of synchronization risk imposes on global investors the task of properly managing this element. Accordingly, global financial agents, such as investment banks, pension funds, mutual funds, and hedge funds, must innovate and develop new financial engineering products that properly help market participants manage this risk.iv.From a broad perspective, our results show the relevance of financial stability within global markets. Well-functioning and stable capital markets are necessary to fulfil the prominent roles of financial markets. Consequently, practioners and portfolio managers ought to incorporate these new perspectives and insights to improve the coordination of the markets before new global shocks again jeopardize the stability of capital markets.

The paper is structured as follows: Section 2 reviews network methods. Section 3 discusses the data, preliminary findings, and applied econometric methods. Section 4 presents the empirical results. Finally, we concludes and extends the findings.

## 2. Materials and Methods

### 2.1. Data

We consider the daily closing prices of fifteen commodities (aluminum, copper, lead, nickel, tin, zinc, gold, platinum, silver, palladium, gasoline, wti, brent, heating oil, and natural gas) and seven exchange rates of strong commodity exporting economies (Chilean peso, Icelandic krona, Norwegian krone, South-African rand, Australian dollar, Canadian dollar, and New Zealand dollar). Additionally, to control for implied volatility effects, we used six implied volatility indices. Specifically, we used Oil 3M index volatility, Copper 3M Index volatility, Gold 3M Index volatility, Aluminum 3M Index volatility, Silver 3M Index Volatility, and CBOE VIX index. The commodity indices’ volatility is based on real-time prices of options on oil, copper, gold, aluminum, and silver, respectively, and is designed to reflect investors’ consensus view of future (3-months) expected commodity price volatility. Similarly, the Cboe Volatility Index^®^ (VIX^®^) is based on real-time prices of options on the S&P 500^®^ Index (SPX) and is designed to reflect investors’ consensus view of future (30-day) expected stock market volatility. These volatility indices are often referred to as the market’s “fear gauge”. We used closing prices to avoid any concern about spurious autocorrelations induced by taking the averages of intra-day prices [19,22,30,31]. Finally, following Magner et al. [32], we included implied volatility indices in our models as part of the out of the sample analysis.

### 2.2. Network Method—The Minimum Spanning Tree Length (MSTL)

We measured commodity market synchronization using the minimum spanning tree length (MSTL). Following the standard procedure described by [24,27,28,32,33,34], we calculated the correlations between commodities and built the asset trees based on commodity returns. The closing commodities prices i at time date τ is $P_{i}\left(\tau \right)$. The return of commodity i is given by $r_{i}\left(\tau \right)=\ln P_{i}\left(\tau \right)-\ln P_{i}\left(\tau -1\right)$, for a consecutive sequence of trading days. For each commodity i daily returns are calculated within a time window of 1 month. Let be $r_{i}^{t}$ the commodity returns vector of the month t, then:(1)$$\rho_{ij}^{t}=\frac{\langle r_{i}^{t}r_{j}^{t}\rangle -\langle r_{i}^{t}\rangle \langle r_{j}^{t}\rangle }{\left[\langle r_{i}^{t2}\rangle -\langle r_{i}^{t}\rangle^{2}\right]\left[\langle r_{j}^{t2}\rangle -\langle r_{j}^{t}\rangle^{2}\right]}$$

$\rho_{ij}^{t}$ is the correlation coefficient between the commodity i and j where 〈I〉 indicates the average over all the trading days of the month t. In this way, a N×N symmetrical matrix $C^{t}$ of correlations between commodities (N is the number of commodities) with values $-1\leq \rho_{ij}\leq 1$.

Then, we converted the correlations of $C^{t}$ in distances $d_{ij}={\left(2\left(1-\rho_{ij}\right)\right)}^{1/2}$ to represent the distance between the commodity i and j. Thus, a correlation $\rho_{ij}=-1$ indicates a maximum distance of $d_{ij}=2$, whilst $\rho_{ij}=1$ indicates a minimum distance of $d_{ij}=0$.

The Minimum Spanning Tree (hereafter, MST) is a tree structure graph that connects the N commodities through N−1 edges avoiding loops and clicks, and where the path to connect all the nodes (in this case commodities) is minimal. The MST was constructed using the Prim algorithm [35]. In this way, MST reduces the information space of the entire network by connecting all nodes with N(N−1)/2 edges, to a tree with N−1 edges. Finally, we estimate the normalized length of the MST (MSTL) as:(2)$$L\left(t\right)=\frac{1}{N-1}\sum_{d_{ij}^{t}\in T^{t}}d_{ij}^{t}$$

The sum of the edges of the resulting tree $T^{t}$ calculated for each month t forms a time series. The variation in the MSTL is calculated as ΔL(t)=lnL(t)−lnL(t−1), which allows us to work with a stationary time series. With this specification, we obtain a dynamic commodities metal tree based on their correlations.

### 2.3. Forecasting Methods

We used two strategies to evaluate the forecasting power of commodity currencies over commodity market synchronization. First, we implemented an in-sample test using time series models (see Table 1 Panel A). Second, to avoid concerns about overfitting, we implemented two out-of-sample tests comparing the benchmark models (Table 1 panel C) with our out-of-sample core models (Table 1 Panel B). Following [32], we built our benchmark models considering autoregressive components of commodity market synchronization and lags of six j implied volatility indices (Oil 3M index volatility, Copper 3M Index volatility, Gold 3M Index volatility, Aluminum 3M Index volatility, Silver 3M Index Volatility, CBOE VIX index). We include these indices as benchmark controls since previous research [32] suggests that implied volatility can be used to predict return synchronization. We employed this benchmark because, in forecasting, the AR(p) models are difficult to beat [36,37], and previous evidence [27] suggests that return synchronization is strongly autocorrelated and persistent, implying that the AR(p) benchmark model is more appropriate than a random walk model.

In Table 1, $\Delta MSTL_{t}$ is the log-difference of commodity market synchronization at time t and we considered three lags of $\Delta MSTL_{t}$ as benchmark controls. $P_{i,t}$ is the exchange rate price of the economy i at time t. Finally, $\varepsilon_{t}$ are error terms. The number of lags considered in our econometrics models (see Table 1) is not economically based but purely empirical.

Table 2 exhibits the correlogram for MSTL and a cross-correlation analysis between MSTL and commodity currencies; as expected, the correlation (in absolute value) rapidly decays with a greater lag. For instance, Table 2 column 2 shows that partial autocorrelation (PAC) is surprisingly high: −0.401. Moreover, the PAC in the second and third lags is −0.142 and 0.053, respectively. Table 2, columns 3–9, suggests a similar result: the cross-correlations between MSTL and each commodity currency rapidly decay with longer lags. Specifically, we chose the first lag of the commodity currencies to simplify our econometrics models, especially for our out-of-sample test.

As mentioned, our purpose was to evaluate the predictive ability of commodity currencies over commodity market synchronization measured by the MSTL. Using the in-the-sample core model (see Table 1 Panel A), we test the null hypothesis $H_{o}:\;\gamma =0$ through a simple t-statistic. This null hypothesis posits that commodity currencies do not have predictive ability over commodity synchronization. We use log returns because financial prices are unit-root processes, and their first differences (log returns) are covariance stationary. In this sense, we require a proper estimation of the long-run variance to apply the central limit theorem; for this reason, we use HAC standard errors. In particular, Newey and West [38] proposed a Barlett kernel to ensure a positive definite variance matrix, and Newey and West [39] considered automatic selection for the lag truncator parameter.

For out-of-sample evaluations, we compare the predictive performance of our out-the-sample core model (see Table 1 Panel B) against our benchmark models (see Table 1 Panel C). The null hypothesis $H_{o}:\;\gamma =0$ indicates that our model reduces to an AR(p). We split our sample into two windows: an estimation window (of size R), and an evaluation window (of size P); of course, T = P + R, where T is the total number of observations. We considered three completely different window sizes to avoid any concerns about overfitting. Rossi and Inoue [40] explained that the results from only one ad hoc window size may still be highly controversial because predictability may be confined to one particular sub-sample and hence not robust to different window sizes. In the first evaluation, we used P/R = 0.2, which was equivalent to estimating our model with 170 observations, and compared the prediction with benchmark models with 34 observations. In the second exercise, we used P/R = 0.4, which employed 145 observations to the out-of-sample core model, and 59 observations to evaluate its forecasting power with respect to benchmark models. In the final window, we used P/R = 0.6, which was equivalent to estimating our model with 128 observations and comparing the prediction with benchmark models with 76 observations.

### 2.4. Out-of-Sample Test

To evaluate the difference between our models in the context of nested models, we used the ENCNEW test proposed by Clark and McCracken [41]. Additionally, following Pincheira and Neumann [42], we implemented a mean directional accuracy test to assess the accuracy of commodity currencies in anticipating the direction change in commodity market synchronization. Since the results of the Wild Clark West test proposed by Pincheira, Hardy, and Muñoz [43] and the ENCt test proposed by Clark and McCracken [41] reached conclusions similar to those of the ENCNEW test, we decided not to report the results of these tests. They can be obtained upon request.

#### 2.4.1. Encompassing Test ENCNEW

For out-of-sample evaluations with nested models, we used the Clark and McCracken ENCNEW test [41]. According to these authors, the asymptotic distribution of nested models is generally non-standard, but the functionals of quadratic Brownian motions depend on how parameters are updated (rolling, recursive, or fixed), the number of excess parameters in the large nesting model, and the P/R limit (where P is the number of observations in the prediction window, and R the number of observations in the estimation window [41,44]:(8)$$ENCNEW=P\frac{P^{-1}\sum_{t=R}^{T-1}{\hat{e}}_{1,t+1}\left({\hat{e}}_{1,t+1}-{\hat{e}}_{2,t+1}\right)}{P^{-1}\sum_{t=R}^{T-1}{\hat{e}}_{2,t+1}^{2}}$$
where ${\hat{e}}_{1}$ is the forecasting error of the benchmark model, ${\hat{e}}_{2}$ is the forecasting error of our proposed model and λ is equal to the limit of R/T, when R and T go to infinity. The asymptotic distribution of the ENCNEW under the null of no predictability was:(9)$$ENCNEW\overset{P,R\to \infty }{\to }\lambda^{-1}\int_{\lambda }^{1}\left\{W\left(s\right)-W\left(s-\lambda \right)\right\}\prime dW\left(s\right),\;\left(Rolling\right)\;$$

Clark and McCracken [41] simulated these integrals of quadratic brownian motions and establish the relevant critical values for different numbers of excess parameters (in our case, k2 = 1), different schemes to estimate our parameters (in our case, one parameter), and different divisions of the database (in our case P/R = 1, 2, and 4). These critical values can be found in their paper.

#### 2.4.2. Mean Directional Accuracy

In this analysis, we examined the accuracy of commodity currencies in anticipating the direction change of commodity market synchronization. In the literature on forecasting, this type of analysis is also frequently used; for example, see [42,45].

Using a test based on the average of the following variable $W_{t}$, we examine the accuracy of our forecasts in estimating whether commodities market synchronization would rise or decrease in the forthcoming month:(10)$$W_{t}=\{\begin{array}{c} 1\\ 0 \end{array}\;if\;\begin{array}{c} \left(\Delta ln\left(MSTL_{t}\right)\right)\left(f_{t-1}\right)>0\\ \left(\Delta ln\left(MSTL_{t}\right)\right)\left(f_{t-1}\right)\;\leq 0 \end{array}$$
where $f_{t-1}$ represents a generic forecast for the one-period change of commodity market synchronization $\Delta ln\left(MSTL_{t}\right)$. Our variable computes a “hit” whenever signals an equivalent movement in $\Delta ln\left(MSTL_{t}\right)$.

## 3. Empirical Findings and Discussion

### 3.1. Descriptive Analysis

In this section, we show a description of our main variables of interest during the period of analysis, namely the MSTL network, MSTL evolution, currencies evolution, commodities evolution, and the temporal association between MSTL and currencies.

#### 3.1.1. Minimum Spanning Tree and Commodity Market Synchronization

We perform a brief comparison of the MSTL networks that comprise base metals, precious metals, and energy commodities during three different periods (See Figure 1); namely, during the Great Financial Crisis or Subprime Crisis (October 2008), during the aftermath of the Subprime Crisis (May 2009), and finally, during the COVID-19 outbreak (February 2020). Our evidence shows that the shapes of the networks change over time. These findings indicate that the synchronization behavior of the network of our sample of commodities is time dynamic and captures the market conditions present in financial markets in those episodes.

#### 3.1.2. Dynamic Behavior of Commodity Market Synchronization

We analyzed the time evolution of commodity market synchronization. Figure 2 exhibits the monthly evolution and the monthly variation of the minimum spanning tree length (MSTL) for the total commodities of our sample for the period October 2005–November 2022. In terms of the behavior of the MSTL, we observe that the time series is dynamic over time, with clear cycles and pathways reaching maximum levels near December 2019 and minimum levels near December 2009. Regarding the monthly variation of the MSTL, we appreciate mild periods of change combined with high fluctuations that broadly range from +30% to −30% monthly. In addition, in terms of the volatility of the monthly variations, the data show two broad periods. The first, with high oscillations near the Great Financial Crisis of 2008–09 and its aftermath, and the second period, with lower volatility variations in the years after.

#### 3.1.3. Commodity Currencies and Commodity Market Synchronization

We studied the time association among commodity currencies and the synchronization of our sample of commodities. Figure 3 depicts a heat map with correlations between the return of commodity currencies and the variation of the MSTL formed by the sample of base metals, precious metals, and energy commodities for October 2005 and November 2022. The black (khaki) color represents the period’s minimum (maximum) values. We can observe the following:(a)There was a positive correlation among the sample of currencies, indicating that they tended to move in the same direction during the analysis period.(b)There was a wide range of correlations among the currencies for the period 2005–2022, with a maximum level equal to 0.85 and a minimum equal to 0.34, indicating heterogeneous reactions to market conditions among the baskets of currencies.(c)The correlation between the variation in the MSTL that groups the sample of commodities and the currencies was negative, suggesting the existence of a negative relationship between the synchronization of the commodities and the variations of the currencies against the USD.

Finally, complementing the above evidence, Figure 4 and Figure 5 exhibit the monthly variations in the returns of the sample of commodities and commodities-currencies for October 2005 and November 2022. Again, the black (khaki) color represents the period’s minimum (maximum) values. As Figure 4 shows, for the case of commodities, the returns change widely over time, with monthly variations in the range of −25% and +15%. Interestingly, the highly negative values coincide with the 2008–9 Great Financial Crisis and the 2020 COVID-19 Outbreak, periods characterized by extreme swings in the financial markets and high levels of uncertainty. Similarly, as Figure 5 depicts, there are significant variations for the commodity currencies, but different from commodities, their behavior shows less monthly variations, in the range of −5% and +15%.

Our preliminary evidence indicates that commodity synchronization is dynamic over time; it is negatively related to commodity currencies and tends to coincide with market turmoil periods observed in financial markets during episodes such as the Great Financial Crisis and the Outbreak of the COVID-19 pandemic.

### 3.2. In-Sample Analysis

Table 3 shows the in-sample baseline results based on the econometric specification from panel A in Table 1. The constant and beta from the autoregressive components were omitted from the report to preserve space, but they are still available upon request. The results provide substantial evidence that a rise in the return of commodity currencies predicts a decline in commodity market synchronization. The dependent variable is the MSTL, and the higher it is, the less synchronized the commodity market is.

In 86% of the exercises, all of the currency return coefficients were positive and significant, ranging from 0.136 to 0.409, with an average of 0.26, implying a robust negative association between commodity currency return and commodity market synchronization. The currencies of Chile, Australia, Canada, Norway, and South Africa always do well with this association, whereas New Zealand and Iceland currencies have a less strong association. These findings are consistent with multiple previous studies that have revealed commodity currencies’ predictability [12,15,18,20,26,46].

Additionally, using the present-value model, Campbell and Shiller [16] explained how exchange rates are determined. Notably, commodities are essential building blocks for these economies, so the causal relationship is expected to go in the direction of commodity prices, dictating the nation’s exchange rate. Nevertheless, the present-value model implies that Granger causality goes the other way, with the country’s currencies forecasting commodity prices, which many studies have empirically supported [12,20,31,46].

Interestingly, even though these countries do not produce all commodities that comprise the synchronization network, we still find that most commodity currencies predict commodity market synchronization. It is noteworthy that all these commodities find widespread use in industry, making it reasonable to assume that their demand will grow as global manufacturing expands. Moreover, they are traded daily and connected through a complex web of trade indices and future contracts.

The main component of Chile’s commodity export bundle, for instance, is copper, which accounts for more than half of the country’s exports and close to 45% of FDI, causing swings in the copper price to influence Chilean peso [12]. Moreover, copper is only 1 of 18 commodities considered in our synchronization bundle, accounting for 5.6% of the commodities used. Nevertheless, our findings demonstrate that Chilean currency can strongly forecast the synchronization of the commodity markets, with the majority of the coefficients significant at 5%.

Our results support a cutting-edge line of research into the extended and broader predictability of commodity currencies and build on earlier findings that some country-specific exchange rates can forecast price fluctuations in their own country’s export bundle of commodities [14,15,16]. Based on this same conjecture, some novel empirical evidence has shown that the exchange rate of nations that are not producers of fuel and some base metals can still be used to forecast the price of these commodities [12].

Evidence from Pincheira and Hardy [12], Pincheira and Hardy [19], and Pincheira, Hardy, and Betancor [20] point to a transitive capacity phenomenon, in which they found that commodity-currencies have a predictive ability over commodity return that extends beyond the commodity bundle that a country produces [12]. Furthermore, our results demonstrate that this predictive ability of commodity currencies can extend to commodity market synchronization, which is very plausible given that commodity prices exhibit strong co-movement [47,48].

Other empirical features of Table 3 merit discussion. As Magner et al. [27] predicted, the MSTL exhibits high persistency; the first three lag delays are statistically significant in all cases at the 1% significance level (to keep things simple, we left these results unreported, but they are available upon request). Next, despite these being predictive relationships, the R2 ranged from 0.33 to 0.367, with an average of 0.326. This finding is significantly higher than what is typically reported in the literature on commodity price prediction. Finally, commodity VIX volatility indices delivered mediocre results, with only 36% of the exercises yielding significant results. However, all significant coefficients are negative, implying that higher idiosyncratic volatility predicts periods of higher commodity market synchronization.

### 3.3. Out-of-Sample Analysis

This section presents two out-of-sample tests: ENCNEW (Table 4) and Mean Directional Accuracy Out-of-Sample Analysis (Table 5).

The in-sample results in Table 3 suggest that a higher commodity currency return can predict a decline in commodity market synchronization. However, Rossi and Inoue [40] criticized in-sample studies for possible data mining and overfitting issues. As a result, we conducted out-of-sample exercises to determine whether these conclusions were held in an out-of-sample scenario.

For out-of-sample evaluations, we compare the predictive performance of our out-of-sample core models (see Table 1 Panel B) against the out-of-sample benchmark models (see Table 1 Panel C). The null hypothesis $H_{o}:\;\gamma =0$ indicates that our model reduces to an AR(p), as depicted in Equation (4) (Table 1 Panel B), or to an AR(p) nested and controlled by implied indices of market volatility (Equation (5) of Table 1, in Panel B). We divided our sample into two windows: estimation (size R) and evaluation (size P); of course, T = P + R, where T is the total number of observations. According to Rossi and Inoue [40], the results from only one ad hoc window size may still be highly controversial because predictability may be confined to one sub-sample and therefore not robust to different window sizes. Therefore, to avoid any concerns about overfitting, we considered three completely different window sizes (P/R = 0.2, 0.4, and 0.6, which correspond to estimating our model with 83%, 71%, and 62% of the sample observations, respectively).

Table 4 reports the results from the ENCNEW test to evaluate the prediction ability of the core models (Table 1 panel B) compared to the benchmark models (Table 1 panel C). Consistent with the in-sample analysis, the core models outperformed the benchmark in 42% of the exercises. For instance, models using Norway’s currency outperformed most benchmark models across all estimation windows, reaching significance in 20 of 21 exercises, followed by models using Canada’s currency, which reached significance in 13 of 21 exercises, and models using Australia’s currency, which reached significance in 10 of 21 exercises. On the other hand, models employing Icelandic currency failed to outperform even a single benchmark model across all estimate window set specifications.

The results point to more significant null hypothesis rejections when considering the models with the largest estimation window. The estimation window set of P/R = 0.4 yielded the most significant results, with significance found in approximately 59% of the exercises, followed by the estimation window set of P/R = 0.2, with significance found in approximately 45% of the exercises, and finally, with the estimation window set of P/R = 0.6, yielding only 22% of significant results across all the exercises.

#### Mean Directional Accuracy

Table 5 shows the success hit rate, or Mean Directional Accuracy (DA), of predictions made using the core models (Equations (4) and (5) of Table 1, panel B). To assess statistical significance, we calculated the hit rate as a simple average of $W_{t}$, where we compare the null hypothesis is $H_{0}:E\left(W_{t}\right)\leq 0.5$ against the alternative hypothesis $H_{1}:E\left(W_{t}\right)>0.5$. To compare the results to a “pure luck” benchmark, we evaluated these hypotheses using the Diebold and Mariano [49] and West [50] test statistic (DMW t-stat). Rejecting the null hypothesis means that our models have a higher success rate than a 50% “pure luck” forecast rate.

The results in Table 5 are impressive because all the models incorporating commodity currencies presented hit rates above 50%. Furthermore, considering all exercises, the average hit rate was 74%, which contrasts markedly with a “pure luck” 50% chance. These results imply that our core models can be a helpful tool for forecasting future changes in the synchronization of commodity markets.

Table 5 contains some additional empirical features worth noting. First, all results were statistically significant at the 1% level. Second, the estimation window specification set of P/R = 0.2 produced the highest average hit rate of 77%, followed by the estimation specification set of P/R = 0.4, which produced a 73% hit rate, and finally, the estimation specification set of P/R = 0.6, which produced a 70% hit rate average. Interestingly, the model estimated until the COVID-19 outbreak (P/R = 0.2) produced the highest average success hit rate, consistent with previous research indicating increased currency predictability during periods of extremely low returns [12].

## 4. Conclusions

This study documents that commodity currencies have a predictive ability on the synchronization behavior of commodity market returns, considering base metals, precious metals, and energy commodities. Our findings support the commodity currency hypothesis and show yet another instance in which currencies of major commodity exporting economies can predict changes in the aspect of commodity prices.

We expand the use of asset networks, which have been widely applied in stock markets, to analyze the synchronization phenomenon of commodities. The methodology of this paper provides an important empirical reference point for the application of network analysis methods and forecasting techniques to study the phenomenon of commodity market synchronization of returns and its impact on systemic risk.

Our research adds to the growing empirical literature on systemic risk derived from the synchronization of returns in financial assets by examining the predictability ability of commodity currencies in the commodity network’s synchronizing behavior. Our findings are useful for practitioners dealing with synchronization risk, which reduces the benefits of diversification while increasing the risk of contagion. Our research shows that considering commodity currencies can be useful when making decisions related to commodity market synchronization risk [14].

Future research could broaden our tests to include other commodities and explore predictions with shorter- and longer-term durations.

## Figures and Tables

**Figure 1 entropy-25-00562-f001:**
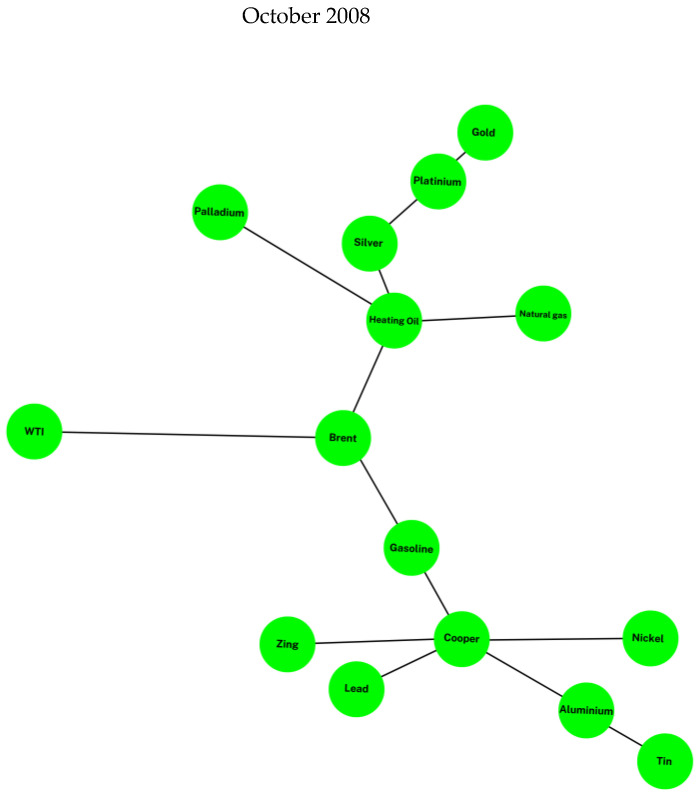
Commodities Minimum Spanning Trees (MST). These figures depict the networks formed by base metals, precious metals, and energy commodities in three different periods of time. The green figure shows the MST in the Subprime Crisis (October 2008), the yellow figure shows the resulting MST in a low volatility period (May 2009), and the gray figure shows the MST during the COVID-19 outbreak (February 2020).

**Figure 2 entropy-25-00562-f002:**
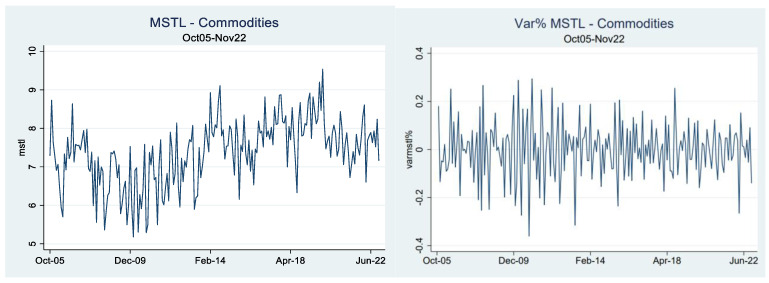
Minimum spanning tree length (MSTL). This figure represents the evolution and variation of the minimum spanning tree length for the sample of commodities for the period Oct2005–Nov2022.

**Figure 3 entropy-25-00562-f003:**
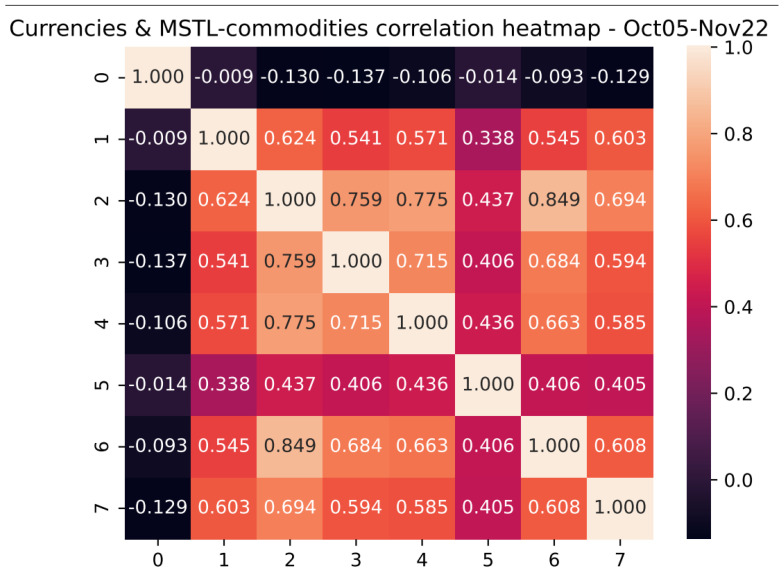
Correlation heatmap. This figure shows the resulting heatmap correlation among the sample of commodity currencies and the MSTL formed by the sample of base metals, precious metals, and energy commodities for the period October 2005 and November 2022. Source: Authors’ elaboration.

**Figure 4 entropy-25-00562-f004:**
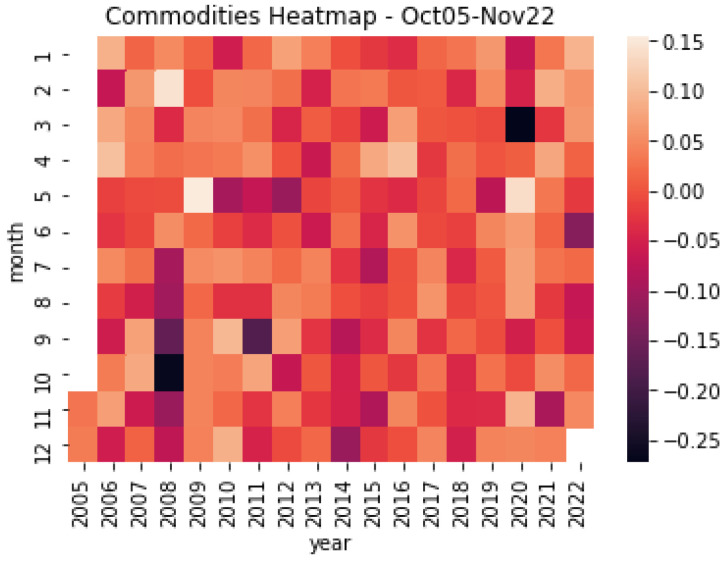
Commodities heatmap. This figure shows the resulting heatmap monthly return of the sample of base metals, precious metals, and energy commodities for the period October 2005 and November 2022. Source: Authors’ elaboration.

**Figure 5 entropy-25-00562-f005:**
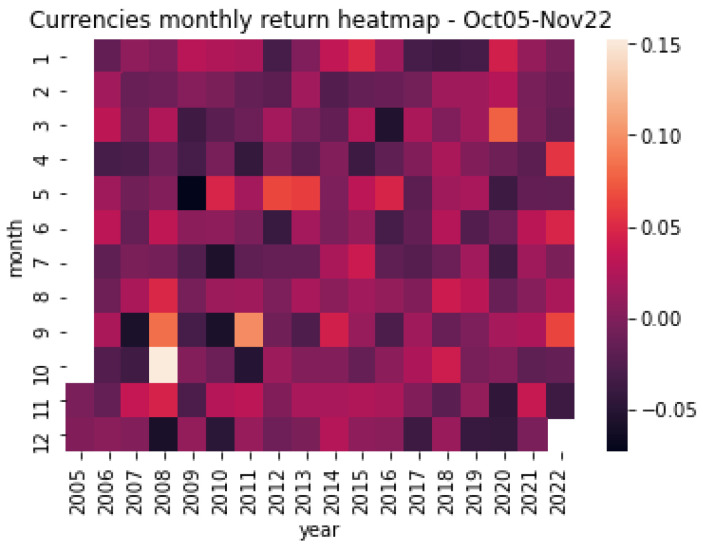
Commodity currencies heatmap. This figure shows the resulting heatmap monthly return of commodity currencies for the period October 2005 and November 2022. Source: Authors’ elaboration.

**Table 1 entropy-25-00562-t001:** Main econometric models for both the core and benchmark models.

Panel A: In-the-sample core model	
(3)	$\Delta ln\left(MSTL_{t}\right)=c+\overset{3}{\underset{n=1}{\sum }}\beta_{n}\Delta ln\left(MSTL_{t-n}\right)+\delta \Delta Vol\_Index_{j,\;t-1}+\gamma \overset{3}{\underset{n=1}{\sum }}\Delta ln\left(P_{i,\;\;t-n}\right)+\varepsilon_{t}$
Panel B: out-of-sample core models	
(4)	$\Delta ln\left(MSTL_{t}\right)=c+\overset{3}{\underset{n=1}{\sum }}\beta_{n}\Delta ln\left(MSTL_{t-n}\right)+\gamma \Delta ln\left(P_{i,\;t-1}\right)+\varepsilon_{t}$
(5)	$\Delta ln\left(MSTL_{t}\right)=c+\overset{3}{\underset{n=1}{\sum }}\beta_{n}\Delta ln\left(MSTL_{t-n}\right)++\delta \Delta Vol\_Index_{j,t-1}+\gamma \Delta ln\left(P_{i,\;t-1}\right)+\varepsilon_{t}$
Panel C: Out-of-sample benchmark models	
(6)	$\Delta ln\left(MSTL_{t}\right)=c+\overset{3}{\underset{n=1}{\sum }}\beta_{n}\Delta ln\left(MSTL_{t-n}\right)+\varepsilon_{t}$
(7)	$\Delta ln\left(MSTL_{t}\right)=c+\overset{3}{\underset{n=1}{\sum }}\beta_{n}\Delta ln\left(MSTL_{t-n}\right)+\delta \Delta Vol\_Index_{j,t-1}+\varepsilon_{t}$

Source: authors’ elaboration.

**Table 2 entropy-25-00562-t002:** Correlogram and cross-correlation analyses.

(1)	(2)	(3)	(4)	(5)	(6)	(7)	(8)	(9)
Lag	MSTL	Chilean Peso	Icelandic Krona	Norwegian Krone	South African Rand	Australian Dollar	Canadian Dollar	New Zealand Dollar
		Panel A: Ret(−1)						
0		−0.009	−0.014	−0.106	−0.129	−0.130	−0.137	−0.093
1	−0.401	0.096	0.133	0.248	0.175	0.233	0.196	0.222
2	−0.142	0.034	−0.090	−0.060	0.063	−0.050	−0.045	−0.105
3	0.053	−0.031	0.075	−0.007	−0.018	0.001	0.019	−0.013
4	0.067	0.032	−0.009	0.028	0.039	0.048	−0.050	0.025
5	−0.050	0.008	0.006	−0.020	−0.029	−0.026	0.064	0.047
6	−0.039	0.029	0.046	0.013	0.017	0.040	0.039	0.005
7	0.062	−0.005	−0.179	−0.036	−0.017	−0.032	−0.068	−0.042
8	−0.118	0.060	0.113	0.071	0.019	0.037	0.020	0.053
9	0.071	−0.141	0.028	−0.089	−0.094	−0.077	−0.015	−0.063
10	0.063	−0.023	−0.026	−0.065	0.037	−0.025	−0.023	−0.036
11	−0.045	0.061	−0.020	0.104	0.030	0.066	0.005	0.103
12	−0.028	−0.047	−0.006	−0.056	−0.018	−0.068	−0.015	−0.182
		Panel B: Ret(−1)+Ret(−2)+Ret(−3)						
0		0.055	0.066	0.109	0.125	0.108	0.102	0.064
1	−0.401	0.019	−0.012	−0.021	0.047	−0.001	−0.049	−0.058
2	−0.142	0.004	0.040	0.002	−0.005	0.015	0.019	0.036
3	0.053	0.042	0.023	0.012	0.015	0.035	0.031	0.049
4	0.067	0.015	−0.067	−0.023	−0.021	−0.011	0.019	0.006
5	−0.050	0.048	−0.009	0.030	0.008	0.026	−0.007	0.010
6	−0.039	−0.051	−0.021	−0.032	−0.049	−0.039	−0.036	−0.035
7	0.062	−0.055	0.058	−0.054	−0.019	−0.041	−0.009	−0.027
8	−0.118	−0.064	−0.009	−0.029	−0.015	−0.019	−0.019	0.001
9	0.071	−0.002	−0.030	−0.012	0.029	−0.017	−0.020	−0.071
10	0.063	0.022	−0.005	0.055	0.002	0.026	0.042	−0.010
11	−0.045	−0.020	0.027	0.003	0.035	0.045	0.034	0.039
12	−0.028	0.009	0.004	−0.003	0.003	0.014	−0.007	0.081

Note: Column 2 shows partial autocorrelations of MSTL. Columns 3–9 show the Pearson correlation coefficients between the contemporaneous MSTL and the lags of each commodity currency. Source: Authors’ elaboration.

**Table 3 entropy-25-00562-t003:** Forecasting commodity market synchronization with commodity currencies: in-sample analysis.

	Australia	Canada	Chile	Iceland	Norway	New Zealand	South Africa
ER(−1) + ER(−2) + ER(−3)	0.233 ***	0.344 *	0.189 **	0.156 ***	0.233 **	0.174	0.190 *
	(0.085)	(0.184)	(0.073)	(0.057)	(0.126)	(0.130)	(0.107)
IVOL Aluminum (−1)	−0.045 **	−0.046 **	−0.042 **	−0.042 **	−0.043	−0.044 **	−0.045 **
	(0.018)	(0.019)	(0.017)	(0.018)	(0.018)	(0.017)	(0.018)
Obs	189	189	189	189	189	189	189
R2	0.356	0.357	0.349	0.349	0.356	0.348	0.355
ER(−1) + ER(−2) + ER(−3)	0.277 ***	0.409 **	0.236 ***	0.130	0.297 **	0.161	0.262 **
	(0.103)	(0.174)	(0.078)	(0.084)	(0.117)	(0.147)	(0.103)
IVOL Cooper (−1)	−0.124 ***	−0.118 ***	−0.112 ***	−0.107 **	−0.115	−0.110 **	−0.120 ***
	(0.045)	(0.041)	(0.041)	(0.042)	(0.042)	(0.045)	(0.045)
Obs	202	202	202	202	202	202	202
R2	0.346	0.347	0.338	0.331	0.347	0.331	0.353
ER(−1) + ER(−2) + ER(−3)	0.214 **	0.337 **	0.186 ***	0.090	0.255 **	0.102	0.223 **
	(0.090)	(0.169)	(0.071)	(0.095)	(0.116)	(0.151)	(0.095)
IVOL Crude Oil (−1)	−0.004 **	−0.003	0.001	0.007	−0.006	0.003	−0.003
	(0.031)	(0.029)	(0.031)	(0.032)	(0.032)	(0.034)	(0.031)
Obs	202	202	202	202	202	202	202
R2	0.324	0.326	0.319	0.314	0.327	0.313	0.332
ER(−1) + ER(−2) + ER(−3)	0.293	0.396 **	0.231 ***	0.136 *	0.301 **	0.171	0.262 ***
	(0.085)	(0.164)	(0.070)	(0.081)	(0.117)	(0.130)	(0.099)
IVOL Gold (−1)	−0.180 ***	−0.165 ***	−0.161 ***	−0.161 ***	−0.167 ***	−0.165 ***	−0.170 ***
	(0.054)	(0.057)	(0.057)	(0.059)	(0.057)	(0.052)	(0.057)
Obs	202	202	202	202	202	202	202
R2	0.362	0.36	0.352	0.345	0.362	0.346	0.367
ER(−1) + ER(−2) + ER(−3)	0.303 ***	0.393 **	0.218 ***	0.135	0.302 **	0.179	0.256 **
	(0.101)	(0.178)	(0.074)	(0.086)	(0.121)	(0.139)	(0.104)
IVOL Silver (−1)	−0.186 ***	−0.166 ***	−0.159 ***	−0.162 ***	−0.169 ***	−0.169 **	−0.168 ***
	(0.055)	(0.050)	(0.050)	(0.051)	(0.053)	(0.050)	(0.049)
Obs	202	202	202	202	202	202	202
R2	0.365	0.336	0.351	0.346	0.362	0.347	0.367
ER(−1) + ER(−2) + ER(−3)	0.236 **	0.351 **	0.206 ***	0.098	0.267 **	0.115	0.232 **
	(0.096)	(0.175)	(0.072)	(0.090)	(0.122)	(0.146)	(0.100)
VIX(−1)	−0.031	−0.024	−0.026	−0.019	−0.027	−0.020	−0.026
	(0.030)	(0.029)	(0.029)	(0.028)	(0.030)	(0.029)	(0.029)
Obs	202	202	202	202	202	202	202
R2	0.327	0.329	0.322	0.315	0.33	0.315	0.334
ER(−1) + ER(−2) + ER(−3)	0.211 ***	0.334 **	0.187 ***	0.092	0.251 **	0.104	0.222 **
	(0.088)	(0.166)	(0.070)	(0.091)	(0.114)	(0.141)	(0.095)
Obs	202	202	202	202	202	202	202
R2	0.324	0.326	0.319	0.313	0.327	0.313	0.332

Note: ER(−1) + ER(−2) + ER(−3) is the cumulative return of the one trimester lag of each commodity currency. The table exhibits estimations of equation presented in Table 1 panel A, with HAC estimators of the long-run variance according to [38,39]. We do not report the constant terms and lags of MSTL for space. Standard deviation is in parentheses. * *p* < 10%, ** *p* < 5%, *** *p* < 1%. Source: Authors’ elaboration.

**Table 4 entropy-25-00562-t004:** Forecasting commodity market synchronization with commodities-currencies: Out-of-sample analysis with the ENCNEW test.

ENCNEW	Australia	Canada	Chile	Iceland	Norway	New Zealand	South Africa
	P/R: 0.2						
AR(3) + IVOLAluminum(−1)	0.261	0.857 **	0.365	−0.025	1.276 **	0.233	0.287
AR(3) + IVOLCCop(−1)	0.563 *	1.275 **	0.822 **	0.100	1.926 ***	0.465	0.720 *
AR(3) + IVOLCrudOil(−1)	0.303	0.900 **	0.344	−0.012	1.340 **	0.274	0.291
AR(3) + IVOLGold(−1)	0.653 *	1.294 **	0.685 *	0.178	2.006 ***	0.565 *	0.969 **
AR(3) + IVOLSilver(−1)	−0.490	0.752 **	0.233	−0.300	0.959 **	−0.107	0.438
AR(3) + VIX(−1)	0.443	0.932 **	0.574 *	0.026	1.501 ***	0.289	0.435
AR(3)	0.144	0.743 *	0.320	−0.019	1.194 **	0.176	0.254
	P/R: 0.4						
AR(3) + IVOLAluminum(−1)	0.992 *	0.982	0.795	0.104	1.656	0.492	1.695
AR(3) + IVOLCCop(−1)	1.658 **	1.427 **	1.332 **	0.363	2.432 **	0.959	2.293 ***
AR(3) + IVOLCrudOil(−1)	0.892 *	0.865 *	0.676	0.039	1.585 **	0.405	1.431 **
AR(3) + IVOLGold(−1)	1.472 **	1.210 **	1.155 **	0.386	2.272 ***	0.882	2.352 ***
AR(3) + IVOLSilver(−1)	0.737 *	0.781 *	0.653	0.104	1.650 **	0.504	1.935 **
AR(3) + VIX(−1)	1.153 **	0.997 *	0.947 *	0.120	1.812 **	0.505	1.682 **
AR(3)	0.923 *	0.896 *	0.744 *	0.091	1.634 **	0.424	1.541 **
	P/R: 0.6						
AR(3) + IVOLAluminum(−1)	0.538	0.475	0.433	−0.165	1.062 *	0.209	0.761
AR(3) + IVOLCCop(−1)	0.385	0.274	0.316	−0.291	1.146 *	0.158	0.368
AR(3) + IVOLCrudOil(−1)	0.385	0.333	0.310	−0.259	0.930 *	0.072	0.449
AR(3) + IVOLGold(−1)	0.650	0.379	0.612	0.056	1.299 *	0.215	1.243 *
AR(3) + IVOLSilver(−1)	−0.215	0.089	0.122	−0.357	0.818 *	−0.308	0.811 *
AR(3) + VIX(−1)	1.024 *	0.726	0.784	−0.101	1.507 **	0.298	1.006 *
AR(3)	0.523	0.459	0.416	−0.201	1.092 *	0.149	0.642

Notes: Each entry reports the EncNew statistic with a recursive schem. The null hypothesis evaluated by the ENCNEW test posits that γ from Equations (4) and (5) in Table 1 Panel B is equal to zero. Then, a rejection of the null hypothesis means that the currency is improving the forecasting accuracy of the model compared to its benchmark. This table compares the predictive ability of our out-of-sample core models (see Table 1 Panel B) with the predictive ability of the benchmark models (see Table 1 panel C). We report three estimation windows with the following critical values: P/R: 0.2 used critical values 0.473, 0.744, and 1.397 for *p* < 10%, *p* < 5%, and *p* < 1%, respectively. P/R: 0.4 used critical values 0.685, 1.079, and 2.098 for *p* < 10%, *p* < 5%, and *p* < 1%, respectively. P/R: 0.6 used critical values 0.791, 1.312, and 2.662 for *p* < 10%, *p* < 5%, and *p* < 1%, respectively. * *p* < 10%, ** *p* < 5%, *** *p* < 1%. Source: Authors’ elaboration.

**Table 5 entropy-25-00562-t005:** Mean Directional Accuracy when Forecasting commodity market synchronization with commodity currencies.

Hit Rate	Australia	Canada	Chile	Iceland	Norway	New Zealand	South Africa
	P/R: 0.2						
AR(3) + IVOLAluminum(−1)	0.765 ***	0.794 ***	0.765 ***	0.735 ***	0.765 ***	0.735 ***	0.794 ***
AR(3) + IVOLCCop(−1)	0.765 ***	0.794 ***	0.735 ***	0.735 ***	0.824 ***	0.765 ***	0.794 ***
AR(3) + IVOLCrudOil(−1)	0.765 ***	0.794 ***	0.765 ***	0.765 ***	0.794 ***	0.765 ***	0.794 ***
AR(3) + IVOLGold(−1)	0.765 ***	0.765 ***	0.765 ***	0.735 ***	0.794 ***	0.765 ***	0.794 ***
AR(3) + IVOLSilver(−1)	0.765 ***	0.765 ***	0.735 ***	0.706 ***	0.765 ***	0.765 ***	0.735 ***
AR(3) + VIX(−1)	0.794 ***	0.824 ***	0.765 ***	0.765 ***	0.824 ***	0.765 ***	0.794 ***
AR(3)	0.794 ***	0.794 ***	0.794 ***	0.765 ***	0.794 ***	0.765 ***	0.794 ***
	P/R: 0.4						
AR(3) + IVOLAluminum(−1)	0.746 ***	0.746 ***	0.746 ***	0.729 ***	0.729 ***	0.729 ***	0.712 ***
AR(3) + IVOLCCop(−1)	0.763 ***	0.729 ***	0.712 ***	0.695 ***	0.780 ***	0.746 ***	0.746 ***
AR(3) + IVOLCrudOil(−1)	0.729 ***	0.763 ***	0.746 ***	0.746 ***	0.763 ***	0.712 ***	0.763 ***
AR(3) + IVOLGold(−1)	0.695 ***	0.729 ***	0.712 ***	0.695 ***	0.763 ***	0.695 ***	0.729 ***
AR(3) + IVOLSilver(−1)	0.712 ***	0.712 ***	0.678 ***	0.695 ***	0.729 ***	0.712 ***	0.729 ***
AR(3) + VIX(−1)	0.746 ***	0.746 ***	0.763 ***	0.729 ***	0.780 ***	0.746 ***	0.763 ***
AR(3)	0.746 ***	0.763 ***	0.746 ***	0.729 ***	0.746 ***	0.729 ***	0.763 ***
	P/R: 0.6						
AR(3) + IVOLAluminum(−1)	0.711 ***	0.684 ***	0.711 ***	0.697 ***	0.697 ***	0.697 ***	0.697 ***
AR(3) + IVOLCCop(−1)	0.697 ***	0.658 ***	0.658 ***	0.658 ***	0.724 ***	0.684 ***	0.711 ***
AR(3) + IVOLCrudOil(−1)	0.697 ***	0.697 ***	0.711 ***	0.724 ***	0.724 ***	0.697 ***	0.737 ***
AR(3) + IVOLGold(−1)	0.658 ***	0.697 ***	0.671 ***	0.671 ***	0.711 ***	0.658 ***	0.697 ***
AR(3) + IVOLSilver(−1)	0.684 ***	0.671 ***	0.658 ***	0.684 ***	0.697 ***	0.684 ***	0.711 ***
AR(3) + VIX(−1)	0.711 ***	0.684 ***	0.724 ***	0.711 ***	0.724 ***	0.724 ***	0.737 ***
AR(3)	0.711 ***	0.697 ***	0.711 ***	0.711 ***	0.711 ***	0.711 ***	0.737 ***

Each entry in this table indicates the rate at which our core models correctly forecast the sign of commodity market synchronization changes. To determine statistical significance, we used the *t*-test of [49,50] against a 50% pure luck benchmark. The HAC standard errors were used and defined based on [38,39]. The star signals represent one-tail significance levels based on the following critical values: 1.28, 1.645, and 2.33 for *p* < 10%, *p* < 5%, and *p* < 1%, respectively. *** *p* < 1%. Source: Authors’ elaboration.

## Data Availability

Data are upon request.

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
