# Peer review of "Forecasting Commodity Market Synchronization with Commodity Currencies: A Network-Based Approach"

_entropy, 2023, doi:10.3390/e25040562_

Round 1
Reviewer 1 Report
I assume the authors are not native English speakers but the paper is well-written. There are just a few instances where more appropriate words or tenses could have been used.
I have made suggestions in the attached copy for the consideration of the authors.

Author Response
First of all, we would like to thank the referee for his/her valuable comments. In the following lines, we address his/her comments/suggestions one by one.
Line 57 – 58. “similarities between the structure of national and regional economies”
Line 74 – 75. “The growing path of cross-commodity correlations and the correlation of commodity prices with prices of other financial asset classes has attracted greater interest.”
Line 146. “which points that”
Line 202. “our results show the relevance”
Line 235. “The closing commodities prices”
Line 264. “time series models”
Line 367. “Descriptive analysis”
Line 538. “these economies”
Line 671 – 672. “of major commodity exporting economies”
Line 667 – 686. “We improved the conclusion based on your suggestions. It is shorter and faster for the reader to comprehend our main takeaways now.”

Reviewer 2 Report
The paper deals with an interesting problem of commodity markets
synchronization, which is measured by the length of the minimum spanning
tree in the correlation matrix of commodity returns (MSTL). The authors
investigate changes in commodity markets synchronization by estimating several econometric models incorporating commodity currencies as well as selected volatility indices and by testing their in-sample and out-of-sample predictive properties. They evidence strong predictive content of commodity currencies.
As regards the presentation, the paper is uneven. It contains a neatly written
introduction with a proper literature review. The empirical investigation is
well designed and executed. However, its description does not correspond in
quality to the introduction part. Although the authors claim numerous practical implication, they are seriously limited by the short forecast horizon (only 1 month ahead) that was investigated. I would suggest toning this down.
Overall, the paper contains a nice contribution to the existing literature, in
my opinion worth to be published.
Specific remarks:
l. 223-225; be specific, what are the implied volatility indices mentioned here
comparing to indices form l. 218-220
l. 23; vigorously growing literature
l. 84, 106-7 and elsewhere; perhaps down-case: copper, aluminum, lead, nickel, tin,zinc
l. 136; this study demonstrates commodity currency predictive
ability
l. 145; points that
l. 192-206; why different font?
l. 194; revisit -> redesign?
l. 268; looking at Table 1, n=1,2,3 is the number of a specific lag, not the total number of lags (which is 3)...
l.270; (see Table 1)
Table 2; not clear why you need Panel B...
l. 326; Brownian
l. 331-4; either explain what is in the formulas (e1, e2, lambda) or just cite Clark and McCracken (2001) with a short summary; write more on how to
interpret the test results either here or in the note below Table 4.
Sec 3.3. Out-of-Sample analysis;
1. In forecasting literature some naive benchmark (eg. random walk) is also
referred to, to see whether there is a gain from using any model at all; perhaps
worth adding here, even in an appendix;
2. There is little motivation for using implied volatilities as control (? they are lagged...) variables and even less discussion on how they impact/differentiate forecasting performance and why.
3. l. 639; P/R=0.6?
Sec. 4. Conclusions; too long and repetitive (compare with Introduction), just briefly summarize your findings, their implications and point out possible following research directions.
Author Response
First of all, we would like to thank the referee for his/her valuable comments. In the attached file we address his/her comments/suggestions one by one.

Round 2
Reviewer 2 Report
1. l. 379-80: this is not a new sentence, start with "where" ; there is no lambda in Eq.4 so mentioning it here is confusing; if you want to define lambda, do it in a separate sentence...
2. l. 669: scheme not schem